# Post-traumatic stress, social, and physical health: A mediation and moderation analysis of Syrian refugees and Jordanians in a border community

**Tara M. Powell**[1]*, **Oe Jin Shin**[1], **Shang-Ju Li**[2], **Yuan Hsiao**[3]

**1** School of Social Work, University of Illinois Urbana-Champaign, Urbana, Illinois, United States of America,
**2** Evaluation Department, Americares, Stamford, Connecticut, United States of America, **3** Department of Sociology, University of Washington, Seattle, Washington, United States of America

* tlpowell@illinois.edu

## Abstract

### Objectives

This study examined the mediating or moderating relationship of social health on physical health and post-traumatic stress symptoms among displaced Syrians and Jordanians at high risk for physical and mental health ailments. Frequency of mental health symptoms stratified by demographic factors was also explored. We hypothesized social health would mediate and/or moderate the relationship between physical and post-traumatic stress symptoms (PTSS).

### Methods

This cross-sectional study includes 598 adults between 18 and 75 years old recruited from three health centers in the city of Irbid, Jordan, 20 km away from the Syrian border. Post-traumatic stress symptoms (PTSS) were measured through the primary care post-traumatic stress disorder checklist. Physical and social health were assessed through the Duke Health Profile. One-way ANOVA and independent samples T-tests examined mean scores of social health, PTSS, physical health stratified by age, gender, nationality, education level, and trauma exposure. Bivariate correlations explored the relationship between social health, PTSS, and physical health. PROCESS macro tested social health as a moderator and mediator on the association of the physical health and PTSS.

### Results

Social health moderated and mediated the relationship between physical health and PTSS. Males reported ($t = 2.53$, $p < .05$) better physical health scores than females. Those who had less than a high school education reported lower social health ($F = 13.83$, $p < .001$); higher PTSS ($F = 5.83$, $p < .001$); and lower physical health ($F = 5.76$, $p < .01$) than more educated individuals. Syrians reported significantly higher PTSS ($F = 4.13$, $p < .05$) than Jordanians, however, there was no significant differences between nationality for physical or

**Data Availability Statement:** Data cannot be shared publicly because it is co-owned by Americares and University of Illinois. Data are available from the University of Illinois Institutional

Data Access / Ethics Committee for researchers who meet the criteria for access to confidential data. The following is the point of contact for the university IRB: Office for the Protection of Research Subjects, Second Floor (MC-095), 805 West Pennsylvania Avenue, Urbana, IL 61801, irb@illinois.edu, 217-333-2670.

**Funding:** Tara M. Powell received the award from Americares Foundation to conduct the study. https://www.americares.org/. Dr. Shang-Ju Li received salary support from the funding organization Americares. Americares participated in preparation of the study, study design, and decision to publish.

**Competing interests:** Shang-Ju Li is an employee of the funding organization, Americares. This does not alter our adherence to PLOS ONE policies on sharing data and materials.

social health. Social health was positively associated with better physical health (r = 0.10, $p < .01$) and negatively with PTSS (r = -.293, $p < .01$).

## Conclusions

Our results support our primary hypothesis suggesting social health mediates and moderates PTSS and physical health. Secondary findings illustrate gender, educational, and income differences in physical health and PTSS.

## Clinical trials registry

NCT03721848

## Introduction

The connection between social, physical, and psychological well-being is well documented. Social support, which is drawn from social networks (e.g., family or friends) has been shown to act as a protective mechanism against psychological distress and to buffer physical health ailments [1–3]. Research has illustrated that social connections may reduce the severity of mental health conditions such as post-traumatic stress disorder (PTSD) and depression and may buffer other psychological distress symptoms [4–6]. Direct links have also been established between physical health ailments and mental health symptomology. For example, PTSD has been associated with a myriad of physical health-related co-morbidities such as chronic pain, poor physical health-related quality of life, and general health complaints [5, 7].

Individuals who live in poverty or have experienced chronic stress and/or traumatic events are at a high risk for physical and mental health ailments [8–11]. Disaster exposed individuals, for example, are at a disproportionate risk of developing co-morbid post-traumatic stress symptoms (PTSS) and vascular problems [12]. While studies have illustrated the linear connection between social, physical, and mental health symptoms, few have examined how social health (e.g. support from social networks and quality of relationships) [13, 14] may play an intermediary role between physical and mental health. To address this, our study examines the influence of social health on the relationship between physical health and post-traumatic stress symptoms in a border community in Jordan with a high population of displaced Syrians, and where the host community experiences high rates of poverty and has been affected by the Syrian refugee crisis over the past several years.

## Background

### Syrian crisis in Jordan

The Syrian crisis has been declared the worst refugee crisis since World War II, displacing over 6 million Syrians to neighboring countries and Europe [15]. According to a 2016 United Nations High Commissioner for Human Rights report, over 5.6 million registered Syrian refugees had fled from their homes and are living in surrounding countries such as Turkey, Lebanon, and Jordan; the number of refugees has grown additional 12.7% from the previous year [16].

Jordan is among the countries with the largest influx of displaced Syrians. As of August, 2020 there were, 658,756 Syrians refugees registered through the United Nations High Commissioner for Refugees (UNHCR), and the Jordanian government estimates approximately 1.4

Syrians remain unregistered in the country [8, 17, 18]. While many of the Syrians in Jordan have experienced the complex trauma of the war, Jordanians have been adversely impacted by the influx of hundreds of thousands of refugees [10, 19]. The inundation of refugees has put a strain on healthcare and social services, and has increased job competition, thereby increasing the potential for civil conflict and societal discord [10, 19, 20]. Given the fragility of Jordan's overloaded system, many low-income Jordanians and displaced Syrians are without health-care, jobs, and social services [19]. Moreover, systems in Jordan have been unable to provide adequate healthcare to individuals with chronic non-communicable diseases (NCDs) (e.g., hypertension, cardiovascular disease, diabetes, chronic respiratory disease) or mental health conditions (e.g., depression, post-traumatic stress, anxiety) [19, 21, 22].

Jordan has also experienced an epidemiological shift from the decreasing incidence of infectious diseases and fertility-related mortality to a rapidly increasing prevalence of chronic NCDs, such as diabetes, obesity, asthma, cardiovascular disease (CVD), and cancer [20, 23]. This transition is due in part to medical advances, which have resulted in reduced mortality and increased ability to prevent fertility-related mortality [24]. These advances have led to rapid population growth, which coupled with globalization, modernization, the refugee crisis, and urbanization, has initiated population level sociocultural changes [23].

In recent years, the global burden of NCDs has gained attention, particularly in low and middle-income countries. It is estimated that 80% of deaths globally are connected to NCDs, which are of particular concern to individuals from middle income countries, such as Syria and Jordan [25]. In Jordan, ischemic heart disease is the leading cause of death (18%), followed by cancer (15%), strokes (12%), and diabetes (7%) [26]. A study of NCDs among Syrians in Jordan found that participants reported at least one family member with an NCD, including hypertension (9.7%), arthritis (6.8%), diabetes (5.3%), chronic respiratory diseases (3.1%), and cardiovascular disease (3.7%) [21].

The global burden of mental health-related disorders has also emerged on the international stage, particularly in LMICs. Common mental health disorders such as anxiety and depression significantly contribute to the overall burden of disease in many LMICs [9]. The National Behavioral Risk Factor Surveillance Survey in Jordan found that 18.4 percent of low-income Jordanians reported frequent mental health distress [27]. Studies have suggested that, among displaced Syrians, clinical levels of post-traumatic stress are as high as 71% [28, 29]. Mental health difficulties in LMICs have been attributed to a myriad of factors, such as food and hous-ing instability, forced displacement, exposure to potentially traumatic events, and the depletion of resources [9, 30].

Chronic diseases and mental health disorders often co-occur. The prevalence of depression in the global population ranges from 3–10%, while up to 33% of individuals with chronic dis-eases experience co-morbid depression [31, 32]. Al-Amer and colleagues [5] found 19.7 per-cent of Jordanians diagnosed with diabetes reported co-occurring depression. Studies of Syrian refugees in Jordan have also found that over half of those with chronic diseases also reported high levels of PTSD, and 35% of Syrians previously diagnosed with a chronic illness experienced clinical rates of depression [12, 33]. The co-morbidity of mental disorders and physical illness accounts for a 37% loss of healthy life years; smoking, alcohol consumption, lack of physical activity, unhealthy diet, and decreased access to healthcare have all been linked to this loss [34]. Co-morbid depression and diabetes, for example, has a slew of negative effects, such as decreases in self-care and self-management, a reduction in adherence to medication, and increases in medical expenditures [35–37].

The primary causes of mortality among individuals with mental health difficulties include preventable conditions such as cardiac illnesses, respiratory and infectious diseases, diabetes, and hypertension [38]. Furthermore, displaced populations often are unable to adequately

treat chronic physical and mental health conditions due to a lack of medications and appropriate health care [1, 33]. Al-Rousan and colleagues [6] found that cost of living, lack of legal work opportunities, and inability to access healthcare were associated with psychological distress and health problems among displaced Syrians in Jordan. Given these startling rates of NCDs and mental health-related issues, scholars have called for continued research to explore those relationships between those social, biological, personal, and economic factors which impact physical and mental health in LMICs [9].

## Social, physical, and mental health

Social health, which is defined as the combination of social networks and social support [39], involves the intersection of a feeling of belonging or acceptance and fulfillment in social roles, which in-turn can boost health and emotional well-being [14, 40, 41]. A growing body of evidence from the past several decades has illustrated causal relationships between role identity (e.g. social ties, self-esteem, social health), longevity, physical, and mental health [3, 42, 43]. A meta-analysis of 148 studies found a significant association between social health and reduced risk for overall mortality due to chronic diseases [44].

Scholars have suggested that social support and networks are comprised of two processes which can influence health and mental health outcomes. The first process involves the provision of informational, emotional, or instrumental resources to an individual to meet his or her physical and emotional needs. The second process, which we will examine in this paper, are the health benefits associated with participation and inclusion in a social group. Social interactions influence emotional, biological, or behavioral responses in an individual. In turn, these social interactions influence a person's sense of personal control, self-concept, and self-worth [3, 40]. Social health may thereby reduce stress during adversity, which can protect individuals from physical health-related difficulties. Likewise, the social factors attributed to physical health outcomes are markedly similar to those that are associated with positive mental health outcomes and therefore should mediate the relationship between physical and mental health [3].

Research has examined the relationship between social, physical, and mental health, however, few scholars have examined the pathways which may influence these correlations [3]. The following study, therefore, sought to explore mediating and moderating role of social health on physical health and post-traumatic stress symptoms (PTSS) within a Jordanian community near the Syrian border. Our primary aim was to examine the mediating and moderating role of social health on the relationship between physical health and PTSS.

## Methods

This cross-sectional study includes the first wave of data in a longitudinal intervention study measuring the impact of a health and mental health awareness program in Jordan. The data included in this study was collected in April 2017 with three health clinics and 598 participants in the city of Irbid, Jordan, 20 km away from the Syrian border. Eligible patients were referred to the study through the treating Ministry of Health physicians at local health centers. Fliers in Arabic were posted with information about the study at the health centers. Individuals were eligible for the study if they were: (1) between 18 and 75 years old; (2) utilized services at the health centers between January and April 2017.

The principal investigator trained the research team in data collection, consenting procedures, confidentiality of data, research ethics, and study methodology. The research team included three Jordanian staff employed by the Royal Health Awareness Society (RHAS) and two Ministry of Health nurses all fluent in Arabic; two United States based Americares staff; and one university researcher. All questionnaires were translated from English into Arabic by

the RHAS research team members, and back translated followed by pilot testing to ensure quality and accuracy of survey measures. The RHAS research team members collected the first wave of data over three weeks via paper surveys.

Individuals who met the inclusion criteria were invited to participate in the study. If the participant was unable to read or write, the consent form was read to them by the research staff collecting the data. Written consent was obtained prior to participation in any study activities. The questionnaire was administered through researcher assisted self-completion to ensure comprehension of the survey items. The Ministry of Health nurses and RHAS staff read the questions in Arabic, repeating and clarifying if the study participants expressed any difficulties with comprehension. Study procedures were approved by the Jordanian Ministry of Health and the first author's institutional review board prior to data collection.

### Measures

**Demographics.** Demographic items included gender, marital status, income, nationality, age, and education.

**Short form of the PTSD Checklist-Civilian Version (PCL-C).** This six-item screener, which was developed for use in primary care, is derived from the PTSD Checklist-Civilian Version to screen for persons with possible PTSS [45]. The items evaluate re-experiencing, avoidance, and hyper-arousal. The full checklist has been validated to screen for PTSD in countries such as China, Armenia, Nepal, and with Arab and Kurdish populations [26, 27, 46, 47]. Questions inquired how often in the past month a person has been bothered by a symptom of PTSD. For example, one question inquired "How often in the past month were you bothered by repeated, disturbing memories, thoughts, or images of a stressful experience." Items were scored on a five-point Likert scale ranging from 1 (not at all) to 5 (extremely). Scores range from 1–30 with 14 being the cutoff for a probable clinical diagnosis of PTSD [45, 48]. Reliability for this sample was adequate, Cronbach's alpha .728.

**Duke Health Profile.** The Duke Health Profile is a 17-item standardized self-report instrument containing six health measures (physical, mental, social, general, perceived health, and self-esteem), and four dysfunction measures (anxiety, depression, pain, and disability). It is a brief tool derived from the Duke UNC Health profile used to measure overall health as an outcome of preventative health and medical interventions. The tool has strong face and convergent validity. Specifically, the discriminant validity separates those with physical health problems and those with mental health problems [49]. This study examined two of the health measures, physical health and social health. The physical health subscale consisted of five items on a three-point Likert scale ranging from 0 (None) to 2 (A lot). Sample questions inquired how often in the past week participants had experienced trouble: (1) walking up a flight of stairs; (2) hurting or aching in any part of the body; (3) getting tired easily. Social health consisted of five questions inquiring about quality of social relationships and how often the participant took part in social activities. Sample questions included how often in the past week the participant: (1) socialized with others; (2) attended social, religious, or recreation activities. A sample of a relationship quality item ask the participant how much they agree with the following statement: (1) I am happy with family relationships. These questions were scored on a three-point Likert scale from 0 (Doesn't describe me at all) to 2 (describes me exactly). To create a composite score for each sub-scale, items are first summed and then multiplied by 10. The range for the social and physical health scales range from 0–100. Reliability in our sample was adequate, with Cronbach's alpha .72 and .65 respectively.

**Brief Trauma Questionnaire (BTQ).** The BTQ is a ten-item self-report questionnaire used to determine if a person has a qualifying life event that meets Diagnostic Statistical

Manual-IV Criteria A for Post-Traumatic Stress Disorder. Items inquire about specific types of traumatic events and are scored by participants marking yes or no to the questions. Scores range from 0 (no traumatic events) to 9 (exposure to 9 traumatic events). Examples of the questions include whether the person has: (1) had a life-threatening injury (2) experienced war or disaster; (3) been in a situation where they were seriously injured [50]. Within our study we further categorized exposure into three categories: 0 (no traumatic experiences), 1 (one traumatic experience), 2 (more than one traumatic experience).

## Statistical analyses

Data were cleaned and descriptive statistics calculated to assess demographic characteristics. A small percentage (1–3%) of the data were missing for our variables of interest (PTSS, social health, physical health), therefore, we excluded the missing data from the analyses. One-way ANOVA and independent samples t-tests examined mean scores of social health, PTSS, and physical health stratified by trauma exposure, age, gender, nationality, and education level. Bivariate correlations examined the relationship between PTSS, social health, and physical health.

To test the moderation we tested the relationships for: (i) the direct effect of the predictor (physical health) on PTSS, (ii) the direct effect of the moderator (social health) on PTSS, and (iii) the interaction effect of the moderator (social health) on physical heath and PTSS [51]. In order to further characterize the moderating effect that social health had on the relationship between physical health and PTSS we used the Johnson–Neymann (J–N) technique. The J-N technique was used to calculate regions of significance and confidence bands to assess the effect of the independent variable across all levels of the moderator variable [49]. The J-N technique identifies points in the range (0–100) of the moderator variable (social health) where the impact of the independent variable (physical health) on the dependent variable (PTSS) is significant and non-significant [52].

The mediation effects were tested in four steps examining the (i) direct effect of social health (mediator) on PTSS (ii) direct effect of physical health (predictor) on social health (mediator), (iii) total effect of physical health (predictor) on PTSS and (iv) significance of the indirect effect using bias-corrected bootstrapping with 95% confidence intervals (CI) [53]. Mediation is proven when the analysis found significant a- and b-paths, as well as a significant indirect effect with 95% confidence intervals that did not include zero [53]. We adjusted for the effect of sociodemographic correlates (age, gender, nationality, trauma exposure, and education level) in the moderation and mediation models. Income was not included as a control variable because most (91.2%) of the sample came from lower income households. All analyses were conducted in SPSS 27.0.

## Results

The study sample was primarily female, 70.1% and Jordanian, 62.1%. Ages ranged from 18–55 + with most participants between 35–54 years old 44.3%. Most of the participants came from relatively low-income households, with 48.8% making less than 200 Jordanian Dinar (approximately 280 U.S. dollars) per month. Only 36.7% of our study sample completed high school or above. Most of the sample (53.4%) experienced at least one traumatic event. The most common trauma experience Syrian participants reported was being in a war zone (73.8%), and among Jordanians was witnessing a violent death (18.6%). See Table 1 for demographic characteristics and Table 2 for percentages of trauma experiences stratified by nationality.

One-way ANOVA and independent sample T-tests examined mean scores of social health, PTSS, physical health stratified by age, gender, nationality, education level, and trauma

**Table 1. Demographic characteristics.**

| N = 598 | Frequency (%) |
|---|---|
| Gender | |
| Female | 419 (70.1) |
| Male | 179 (29.9) |
| Missing | 0 (0.0) |
| Age | |
| 18–34 | 113 (18.8) |
| 35–54 | 286 (47.8) |
| 55+ | 143 (23.9) |
| Missing | 56 (9.3) |
| Nationality | |
| Jordanian | 371 (62.1) |
| Syrian | 223 (37.4) |
| Egyptian | 2 (0.3) |
| Palestinian | 1 (.2) |
| Missing | 1 (.2) |
| Monthly Income | |
| Less than 200 JD | 284 (48.8) |
| 200–500 | 247 (42.4) |
| 500–1,000 | 51 (8.7) |
| Missing | 16 (2.6) |
| Marital status | |
| Married | 491 (85.1) |
| Other | 86 (14.9) |
| Missing | 21 (3.5) |
| Education | |
| Less than High School | 367 (63.3) |
| High school | 126 (21.7) |
| Above High school | 87 (15.0) |
| Missing | 18 (3.0) |
| Number of traumatic experiences | |
| None | 237 (46.6) |
| One | 112 (22.0) |
| More than one | 160 (31.4) |
| Missing | 89 (14.8) |

exposure. Those who were above 55 years old had lower self-reported physical health $F(2, 512)$ = 7.47, p < .01 than younger individuals. Males reported significantly $t(591)$ = 2.53, p < .05 better physical health scores than females. Those who had less than a high school education reported significantly lower social health $F(2, 576)$ = 13.83, p < .001; higher PTSS $F(2, 569)$ = 5.83 p < .001; and lower physical health $F(2, 574)$ = 13.03, p < .001 than more educated individuals. Those with lower income (<200JD per month) scored lower on social health $F(2, 578)$ = 6.86, p < .01; lower physical health $F(2, 576)$ = 5.76, p < .01; and higher PTSS $F(2, 571)$ = 14.48, p < .001. Syrians reported significantly higher PTSS scores than Jordanians $F(2, 584)$ = 4.13, p < .05. There were no significant differences between Syrians and Jordanians on social or physical health. Table 3 presents t-tests and ANOVA results.

Bivariate correlations indicated that social health was significantly positively associated with better physical health ($r$ = 0.10, $p$ < .05) and negatively associated with PTSS ($r$ = -.22,

**Table 2. Trauma experience type by nationality.**

| Trauma type | Syrian | Jordanian |
|---|---|---|
| War zone | 153 (73.8%) | 35 (9.9%) |
| Serious car accident | 54 (25.8%) | 57 (16.2%) |
| Natural or technological disaster | 35 (16.8%) | 17 (4.8%) |
| Life threatening illness | 23 (11.1%) | 20 (5.6%) |
| Physical Attack | 34 (16.3%) | 25 (6.9%) |
| Violent death of family member | 63 (30.1%) | 68 (18.6%) |
| Witnessed death of someone | 71 (33.6%) | 52 (14.2%) |
| Other experience | 48 (22.9%) | 26 (7.2%) |

$p < .01$). Higher PTSS was negatively associated with better physical health ($r = -0.35$, $p < .01$) and social health ($r = -0.293$, p < .01). Table 4 presents correlation's between post-traumatic stress symptoms, physical health, and social health.

**Table 3. *T*-test and one-way ANOVA of demographic differences in social health, Post-Traumatic Stress Symptoms (PTSS), and physical health.**

| | Social health | | | PTSS | | | Physical health | | |
|---|---|---|---|---|---|---|---|---|---|
| | M (SD) | t | Cohen's d | M (SD) | t | Cohen's d | M (SD) | t | Cohen's d |
| **Gender** | | | | | | | | | |
| Male | 61.58 (20.25) | 2.52* | 0.29 | 14.30 (5.67) | -1.63 | -0.09 | 46.31 (25.89) | 2.53* | 0.21 |
| Female | 57.13 (18.27) | | | 15.18 (6.11) | | | 40.62 (24.61) | | |
| **Marital status** | | | | | | | | | |
| Married | 58.85 (19.06) | -0.06 | 0.01 | 14.65 (5.87) | 2.59* | 0.22 | 42.39 (24.89) | -0.05 | .06 |
| Other | 58.72 (17.67) | | | 16.48 (6.47) | | | 40.81 (25.68) | | |
| **Income** | | F | η² | | F | η² | | F | η² |
| Less than 200 JD | 49.18 (18.14) | 6.96** | 0.03 | 16.14 (6.28) | 14.48*** | 0.06 | 39.28 (24.06) | 5.76** | 0.02 |
| 200–500 | 54.30 (17.31) | | | 13.84 (5.65) | | | 44.26 (25.59) | | |
| More than 500 JD | 55.88 (16.02) | | | 12.31 (5.28) | | | 50.78 (36.06) | | |
| **Nationality** | | | | | | | | | |
| Jordanian | 59.38 (19.03) | 1.23 | 0.04 | 14.43 (6.27) | 4.13* | 0.01 | 43.76 (25.58) | 1.81 | 0.01 |
| Syrian | 56.97 (18.84) | | | 15.87 (6.89) | | | 39.91 (24.33) | | |
| Other | 53.33 (25.17) | | | 10.33 (2.08) | | | 33.33 (11.55) | | |
| **Age** | | | | | | | | | |
| 18–34 | 56.64 (17.62) | 1.07 | 0.01 | 14.32 (5.91) | .66 | 0.001 | 49.38 (23.76) | 7.47** | 0.03 |
| 35–54 | 59.39 (17.99) | | | 14.75 (6.66) | | | 43.08 (24.88) | | |
| 55+ | 59.78 (17.26) | | | 15.28 (7.05) | | | 37.10 (26.23) | | |
| **Education** | | | | | | | | | |
| Less high school | 55.47 (18.32) | 13.83*** | 0.05 | 15.67 (6.30) | 5.83** | 0.04 | 38.18 (25.01) | 13.03*** | 0.04 |
| High school | 63.33 (20.79) | | | 13.64 (5.59) | | | 47.78 (23.56) | | |
| Above high school | 64.71 (16.83) | | | 13.94 (8.18) | | | 50.57 (24.94) | | |
| **Number of traumatic experiences** | | | | | | | | | |
| None | 59.49 (18.66) | 4.757** | 0.02 | 13.99 (6.27) | 15.03*** | 0.06 | 44.56 (25.17) | 7.09** | 0.03 |
| One | 61.96 (16.97) | | | 14.06 (4.79) | | | 44.38 (24.67) | | |
| More than one | 55.25 (19.20) | | | 17.11 (5.87) | | | 41.73 (24.83) | | |

* < .05

** < .01

*** < .001.

**Table 4. Correlations (Pearson's *r*) between post-traumatic stress symptoms, physical health, and social health.**

| Variables | 1 | 2 | 3 |
|---|---|---|---|
| 1. Social health | - | 0.10* | -0.22*** |
| 2. Physical health | - | - | -0.35*** |
| 3. PTSS | - | - | - |

\* < .05, \*\* < .01

\*\*\* < .001.

## The moderation and mediational role of social health on the relationship between PTSS and physical health

Moderation analyses illustrated that higher physical health (*B* = -0.14, SE = 0.05, p<0.001) and social health (*B* = -0.13, SE = 0.03, p<0.001) were significantly associated with lower PTSS scores. The positive coefficient of the interaction term of physical health and social health indicated that social health was a significant moderator on the relationship between self-reported physical health and PTSS (*B* = 0.001, SE = 0.001, p<0.5). Table 5 presents outcomes of moderation analysis; Fig 1 visualizes the moderation effect).

Results from the J-N technique indicated the conditional effect of physical health on PTSD transitioned in significance when the summed score of social health was 61; b = -0.04, SE = 0.02, t = -1.97, p = .05, 95% CIs [-0.07, .00]. The relation between physical health and

**Table 5. Moderation analysis testing social health as a moderator between the relationships of physical health and post-traumatic stress symptoms.**

| Variable | *B* | *SE B* | LLCI | ULCI | *p* |
|---|---|---|---|---|---|
| Gender (ref: male) | | | | | |
| Female | -0.22 | 0.62 | -1.45 | 1.00 | 0.72 |
| Age (ref: 18–34) | | | | | |
| 35–54 | 0.35 | 0.73 | -1.08 | 1.78 | 0.63 |
| 55+ | -0.03 | 0.84 | -1.68 | 1.63 | 0.98 |
| Nationality (ref: Jordanian) | | | | | |
| Syrian | 0.28 | 0.68 | -1.07 | 1.58 | 0.26 |
| Other | -2.63 | 5.53 | -13.51 | 8.25 | 0.70 |
| Education (ref: less than high school) | | | | | |
| High school | -0.36 | 0.37 | -1.08 | 0.37 | 0.63 |
| Above high school | -0.39 | 0.28 | -0.94 | 0.15 | 0.98 |
| Marital status (ref: other) | | | | | |
| Married | -0.59 | 0.79 | -2.13 | 0.96 | 0.46 |
| Number of trauma (ref: none) | | | | | |
| One time | 0.24 | 0.73 | -1.19 | 1.67 | 0.74 |
| More than one time | 1.83* | 0.73 | 0.40 | 3.27 | 0.01 |
| Physical health | -0.14*** | 0.05 | -0.22 | -0.06 | 0.000 |
| Social health | -0.13*** | 0.03 | -0.19 | -0.07 | 0.000 |
| Interaction (Physical health x Social health) | 0.001* | 0.001 | 0.0001 | 0.003 | 0.04 |
| Intercept | 25.28 | | | | |
| *R²* | 0.21 | | | | |
| F | 8.04*** | | | | |

\* < .05, \*\* < .01

\*\*\* < .001.

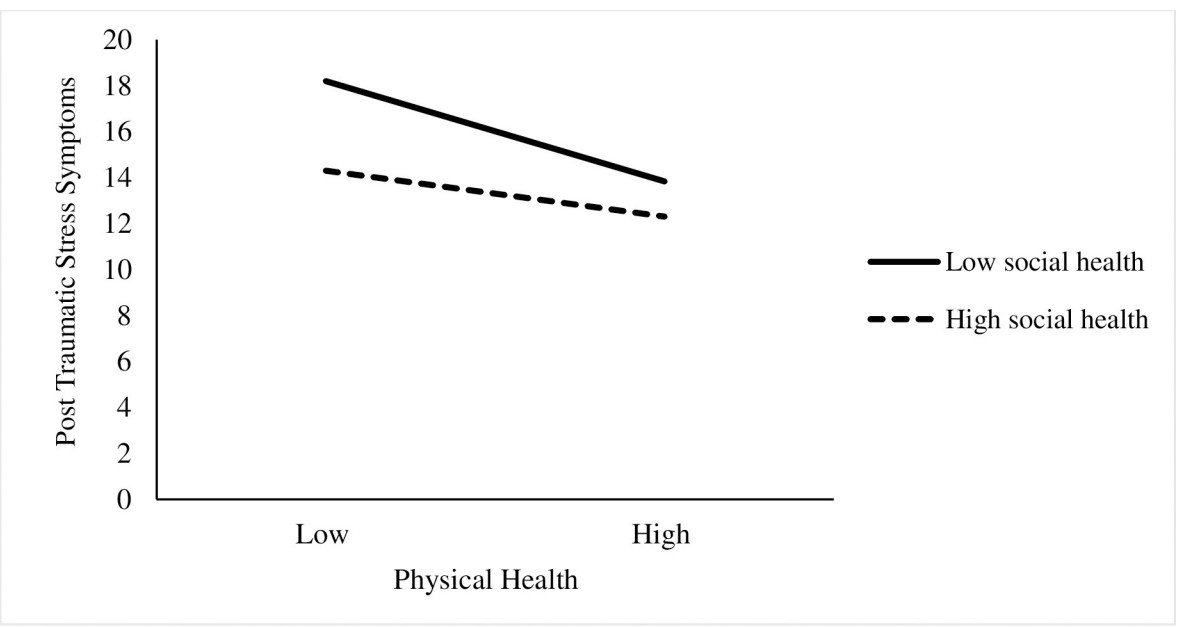

**Fig 1. Relationship between physical health and post-traumatic stress symptoms at low and high levels of social health.**

PTSS scores were significant when social health scores were below a score of 61 and non-significant with higher values of social health (61–100).

Step one of the mediation analysis demonstrated a negative and significant direct effect of social health on PTSS, b = -0.07, s.e. = .015, p = < .001, indicating those with lower social health scored higher on PTSS. Step 2 found that the direct effect of physical health on social health was positive and significant, b = .10, s.e. = .03, p < .01, indicating higher physical health scores were associated with higher social health. Step three indicated the total effect of physical health on PTSS was negative and significant, b = -0.07, s.e. = .011, p = < .001, demonstrating those with higher self-reported physical health scored lower on PTSS. Step 4 demonstrated the indirect effect of physical health on PTSS via social health was statistically significant [β = -.0083, 95% C.I. (-.0147, -.0016)]. Results therefore indicated a negative predictive relationship between physical health and PTSS as mediated through social health. See Fig 2 for a visualization of the mediated model.

## Discussion

The primary aim of our study was to explore the mediating and moderating effect of social health on the relationship between PTSS and physical health. Numerous studies have illustrated the linear connection between social health, physical health, and PTSS however, few have examined whether social health plays a moderating or mediating role. Our findings indicated that social health mediated and moderated the association between physical health and PTSS, demonstrating that the relationship between physical health and PTSS may depend on an individual's level of social health. It is likely this relationship exists because individuals who experience high PTSS and poor physical health may be less likely to participate in activities where social connections are present. However, greater social connections may create a positive influence on an individual's psychological and physical well-being, thereby disrupting the relationship between PTS symptomology and poor physical health.

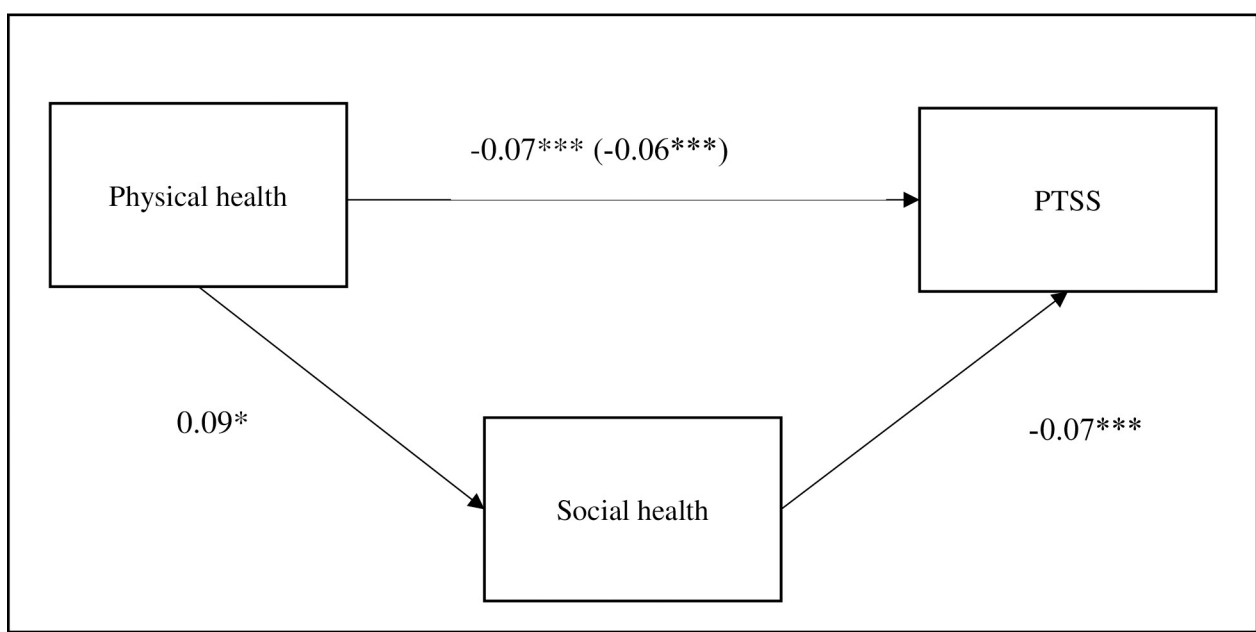

**Fig 2. Mediation model of social health in relationship between physical health and post-traumatic stress symptoms.** Values presented are standardized regression coefficients. The values in parentheses represents the coefficient for direct (without mediator) path.

Consistent with past research, the present study found that lower social and physical health scores were significantly associated with higher PTSS [7, 54–56]. As previously demonstrated, individuals with high levels of social support are less likely to develop clinical levels PTSS than those who reported lower levels of support [57, 58]. Further, social support has also been linked to PTSS severity; those with lower levels of support report more severe and chronic symptoms. Similarly, research has illustrated that higher PTSS have a relationship with greater physical health difficulties [6, 7, 59]. Our study results, therefore, add to the extensive literature connecting PTSS, social, and physical health.

We also found that higher social health was related to greater physical health. The causal relationship between social and physical health is well documented [60, 61]. Research has also indicated that social relationships have a direct short and long-term impact on health, even when controlling for other variables that may influence mortality such as health behaviors and socioeconomic status [60, 62]. Our findings contribute to the body of research demonstrating the relationship between physical and social health.

Interestingly, our study found no significant differences between Syrians and Jordanians on social or physical health, although the Syrians did score significantly higher on PTSS. The mean PTSS scores, however, for both Jordanians and Syrians was above the clinical cutoff of 14 for probable diagnosis of PTSD [48]. Considering the Jordanian sample had low education and experienced high rates of poverty, it is not surprising that significant differences on physical and mental health would not exist. It is well documented that those with lower educational attainment from socio-economically disadvantaged backgrounds in areas directly or indirectly affected by conflict tend to have worse physical health and higher mental health-related symptoms [9, 30, 63, 64].

We also found that those with lower income scored significantly higher on PTSS and lower on social and physical health scores than wealthier respondents. This is not unexpected as many individuals from lower socio-economic backgrounds are at a higher risk for trauma exposure, have greater likelihood of housing instability and less access to resources such as

medical care [65, 66]. As scholars have noted, the highest rates of PTSS in lower-income countries are often attributed to sociodemographic factors and trauma exposure where there is a lack of trained professionals and access to mental health services is low [9, 67].

Gender also appeared to influence physical health; males reported significantly higher physical health scores than females. This finding supports previous research that women, specifically in Jordan, are more likely to report lower physical health than men [68]. Other studies have indicated that inactivity, which can lead to poorer physical health, is higher among women than men in Arab culture [69]. Gender norms such as women needing to be chaperoned in public areas, conservative attire that is not appropriate for physical activity, and the lack of gender separate fitness centers have all been attributed to lower physical activity among women in countries such as Jordan [69–71]. A study examining Syrian women in Jordan found that displacement challenges such as access to healthcare, unemployment, financial difficulties, and lack of support systems are also contributing factors to compromised physical and mental health [2].

Surprisingly, we did not find significant gender differences on PTSS scores. Females did score slightly higher on PTSS than male respondents, however, the differences were not as substantial as previous studies examining gender differences in PTSD [72–74]. While research has consistently found that females report higher levels of PTSS, most of the studies have been conducted in non-Arab settings, therefore, cultural factors may have influenced our findings. Additionally, much of the research examining gender differences in PTSD have employed extensive clinical measures [74]. Our study, however, used the PCL-C, which is a screener for PTSS, but not a tool for clinical diagnoses.

## Implications for policy and practice

Results from this study have implications for interventions designed to prevent mental and physical health related difficulties. Mental health support and NCD prevention in primary care is a growing field given the high rates of co-morbidity. The body of research in this area, specifically in LMICs and humanitarian settings, has significantly grown in the last decade, yet more research is needed to understand appropriate pathways to reduce mental health difficulties and manage or prevent physical health ailments. This is particularly relevant in LMICs with a large influx of displaced people, where much of the population is below the poverty level and without access to physical or mental health care [75]. The World Health Organization has recognized the global burden of NCDs and mental health and advocated for both in the international health agenda [76–78]. Therefore, our findings may inform NCD and mental health interventions in LMICs and humanitarian populations [79]. As Thoits [3] noted, it is crucial to understand intervening mechanisms such as social health on physical or mental health outcomes to design effective interventions.

Considering the high rate NCDs and mental health symptoms in this region, intervention approaches at the community level could be designed to increase individuals physical, social, and emotional well-being [30]. Such approaches may include integration of psychoeducation group work into the primary care setting informing individuals on common reactions to stress and trauma, the connection between post-traumatic stress and physical health, or on stress management. While this type of information does not require clinical mental health services, it may be a strategic way to reach those at risk for mental health-related symptoms. Activities which increase social interaction and inclusion could also be beneficial in health clinics and community centers. For example, activities for greater social contact could include religious events, participating in sporting events, or having community-based health and mental health education [80, 81].

## Limitations

While this study provides novel findings on the moderating role of social health on physical health and PTSS, results should be considered in light of several limitations. First, our study is not generalizable to the wider population. The study was conducted in a region where many people (both Jordanian and Syrian) had high levels of PTSS. In the general global population, clinical rates of PTSS are much lower (approximately 3%) [82], therefore, additional studies in lower-risk populations would generalize the findings. Additionally, we used a self-reported measure of physical health. A more objective health measure, such as blood pressure, body mass index, or fasting blood glucose could further examine whether social health may influence prevention or management of specific NCDs such as hypertension or diabetes. It is also important to note that the reliability of the social health scale was acceptable (.65), however, future studies may want to employ a measure of social health designed specifically for an Arab population, which may yield higher reliability.

We also were unable to identify selection bias because we do not have information on how many people were approached by their doctor to participate in the study, who opted into the study from viewing the fliers that were posted in the health centers, and who fit the inclusion criteria, but opted out of the study. Finally, the data used for this study was cross-sectional, therefore longitudinal causal relationships cannot be determined.

## Conclusion

The interplay of physical and mental health is complex and multifaceted. It is therefore crucial to understand the mechanisms which may influence or act as a buffer for adverse health and mental health outcomes. As the field is continuing to explore the interrelationship between physical health, PTSS, and social health our study contributes to the growing body of research on this area of inquiry. Findings will inform future research examining traumatic stress-related mental health conditions and factors such as social health that may not only buffer those symptoms but also moderate physical health and PTSS. This is particularly relevant in LMICs such as Jordan that have absorbed hundreds of thousands of refugees over the past decade. While humanitarian responses have focused on mental and physical health-related conditions independently, the connection between physical and mental health has been well established. Studies should continue to examine the interrelationship of mental, physical and social health of displaced and other high-risk populations.

## Acknowledgments

The Royal Health Awareness Society, Americares, Aseel Farraj, Mohammad Ayyad, Dr. Rami Farraj, Mariam Abdoh

## Author Contributions

**Conceptualization:** Tara M. Powell, Shang-Ju Li.

**Data curation:** Tara M. Powell, Oe Jin Shin, Shang-Ju Li, Yuan Hsiao.

**Formal analysis:** Tara M. Powell, Oe Jin Shin, Yuan Hsiao.

**Funding acquisition:** Tara M. Powell.

**Investigation:** Tara M. Powell.

**Methodology:** Tara M. Powell, Oe Jin Shin, Shang-Ju Li.

**Software:** Tara M. Powell.

**Validation:** Tara M. Powell.

**Visualization:** Tara M. Powell, Shang-Ju Li.

**Writing – original draft:** Tara M. Powell, Oe Jin Shin, Shang-Ju Li.

**Writing – review & editing:** Tara M. Powell, Oe Jin Shin, Shang-Ju Li, Yuan Hsiao.

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
