## [Decision Letter · Decision Letter 0]

20 Jul 2020

PONE-D-20-11520

Post-traumatic stress, social, and physical health: A mediation and moderation analysis of Jordanians and Syrian refugees in border communities

PLOS ONE

Dear Dr. Powell,

Thank you for submitting your manuscript to PLOS ONE. After careful consideration, we feel that it has merit but does not fully meet PLOS ONE’s publication criteria as it currently stands. Therefore, we invite you to submit a revised version of the manuscript that addresses the points raised during the review process.

We look forward to receiving your revised manuscript.

Kind regards,

Neil J. Vincent, Ph.D.

Academic Editor

PLOS ONE

Journal Requirements:

2. During our internal checks, the in-house editorial staff noted that you conducted research or obtained samples in another country. Please check the relevant national regulations and laws applying to foreign researchers and state whether you obtained the required permits and approvals. Please address this in your ethics statement in both the manuscript and submission information.

3. Thank you for including your competing interests statement; "I have read the journal's policy and the authors of this manuscript have the following competing interests: Shang-Ju Li is an employee of the funding organization, Americares"

4. 

We note that you have indicated that data from this study are available upon request. PLOS only allows data to be available upon request if there are legal or ethical restrictions on sharing data publicly. For more information on unacceptable data access restrictions, please see http://journals.plos.org/plosone/s/data-availability#loc-unacceptable-data-access-restrictions.

5. 

Please include your tables as part of your main manuscript and remove the individual files. Please note that supplementary tables (should remain/ be uploaded) as separate "supporting information" files

Reviewers' comments:

Reviewer's Responses to Questions

**Comments to the Author**

1. Is the manuscript technically sound, and do the data support the conclusions?

Reviewer #1: Yes

2. Has the statistical analysis been performed appropriately and rigorously? 

Reviewer #1: Yes

3. Have the authors made all data underlying the findings in their manuscript fully available?

Reviewer #1: No

4. Is the manuscript presented in an intelligible fashion and written in standard English?

Reviewer #1: Yes

5. Review Comments to the Author

Reviewer #1: Thank you for submitting your manuscript to PLOS ONE. The study holds valuable information and advancement in the understanding of Syrian refugees and host community populations’ physical and mental health in the Jordan.

The following are my comments:

In general, the manuscript would benefit from English professional editing. There are many places with spelling issues, grammatical issues and other issues that can be addressed.

Abstract:

The abstract need to include the results of the difference between Syrian refugees and local Jordanians at high risk of physical and mental health ailments. Such comparison needs to be also included in the results section, since it is well established that Syrian refugees have many physical and mental health issues due to experiencing multiple traumatic events of the war. However, Jordanians have not experienced such events and therefore such comparison is required.

Background:

The title needs to be changed. It is the Syrian Crisis and its impacts in Jordan, or Syrian refugees in Jordan.

In the second paragraph, it is important to state that the number of Syrian refugees registered are by UNHCR in Jordan. Jordan has hosted more than 1.4 million Syrian refugees, who are not registered by UNHCR.

Page 4: Please write the full term of PTS, since it can be confused with PTSD.

Page 5: Forced migration, please change the tem to forced displacement, since migration is associated with a choice.

Page 6: the authors say “… which may be due to a lack of medications and appropriate health care, coupled with high risk for mental health issues that is caused by trauma these individuals may have experienced in their country of origin (33)”. Syrian refugees suffer from extreme poverty and insufficient resources that prevent them from seeking care. In Jordan, health care is private and very expensive. These consideration needs to be taken into account when considering the reasons to why Syrian refugees have high prevalence rates of NCDs.

Please consider including the following in the background literature review and references:

Al-Rousan T, Schwabkey Z, Jirmanus L, Nelson BD. Health needs and priorities of Syrian refugees in camps and urban settings in Jordan: perspectives of refugees and health care providers. East Mediterr Heal J. 2018; 24(3):243–53. Available from: https://doi.org/10.26719/2018.24.3.243

Ibraheem B Al, Kira IA, Aljakoub J, Ibraheem A Al. The health effect of the Syrian conflict on IDPs and refugees. Peace Confl. 2017; 23(2):140–52. Available from: http://doi.apa.org/getdoi.cfm?doi=10.1037/pac0000247

Rizkalla N, Segal SP. Well-being and posttraumatic growth among Syrian refugees in Jordan. J Trauma Stress. 2018; 31(2):213–22. Available from: http://doi.wiley.com/10.1002/jts.22281

Rizkalla et al. Women in refuge: Syrian women voicing health sequelae due to war traumatic experiences and displacement challenges. Journal of Psychosomatic Research. 2020; https://doi.org/10.1016/j.jpsychores.2019.109909

Social, physical, and mental health

Please start this paragraph with the definition of social health.

Please include the hostility among Jordanian towards Syrian refugees and the lack of work permit by the Jordanian government, which limits the income of refugees and their social interactions and may therefore have a negative impact on their self-control, self-concept, and self-worth. You may consider this reference as well:

Achilli L. Syrian refugees in Jordan: a reality check. Migr Policy Cent Eur Univ Inst. 2015; 2(February):1–12. Available from: http://www.reach-initiative.org/wp-content/

The background is a bit confusing, since it is focuses on the Syrian refugees and their physical and mental health. However, the authors states in page 7 that their study aims at examining the physical and mental health “within a Jordanian community near the Syrian border”. Where is the literature review related to the Jordanian community? The statistic of physical and mental health issue in Jordan is completely different than the one related to the Syrian refugees!!!

Methods

Page 7: Please state that the fliers with information on the study were in Arabic.

Page 8: Who were the research team trained by the PI? Were they Arabic speakers? Jordanians? Syrians? Both? Please specify their educational background as well since they may have affected the responses of participants.

Page 8: “self- report questionnaires” means that the participants have filled out the survey independently. However, according to the explanation provided, the research team helped participants with the survey. Did they interview them or only answered questions when needed?

Please add ethical issues related to anonymous participation, confidentiality, incentives if provided, oral consent form or written consent form, etc.

Page 9: Reliability of social health scale is low. Asking refugees who hardly find the means to eat or for transportation to reach their illegal work places about “attended social, religious, or recreation activities” is problematic. Please include this as part of the study limitations.

Page 10: Usually the measures section starts with the description of demographics. Please relocate.

Page 11: How was the mediation effects tested? Were they also done via SPSS PROCESS? Please clarify.

Page 11: “We adjusted for the effect of sociodemographic correlates (age, gender, nationality, trauma exposure, and education level) in the moderation and mediation models.” Income needs to also be included with the controlled variables. If the authors found that all participants were low on income, they need to indicate that this was the reason to not include this variable as one of the controlled variables.

Page 12: Please indicate results related to the comparison between Jordanians and Syrians, other than the obvious “Syrians reported significantly higher PTS scores than Jordanians”. If there were no other differences, please indicate that. I also think that all the analyses need to be divided into Jordanians and Syrians to see the difference between the two groups, especially because of the large sample size. Also, who are the “others” in the sample? Are they Palestinians?

Discussion

Though the findings are important in the field of trauma studies, they lack addressing the specific population. Syrian refugees who were uprooted and displaced, naturally lack the social connections and interactions that Jordanians have. Therefore, the whole analyses need to address these differences both in the results section and the discussion. The authors only mention this comparison in page 14. Please re-write the discussion to put these results upfront.

Also, please add into the results and discussion what were the main traumatic events experienced by Jordanians, since for the Syrians, the war events are obvious.

Page 15: Gender differences in PTSD among Syrian refugees can also be found in: Rizkalla N, Segal SP. War can harm intimacy: consequences for refugees who escaped Syria. J Glob Health. 2019; 9(2):1–10. Available from: https://doi.org/10.7189/jogh.09.020407

Page 17: “For example, activities for greater social contact could include religious events, participating in sporting events, or having community-based health and mental health education”. Please consider adding a budget for transportation or allocating buses/taxes, especially with activities related to Syrian refugees, so they can actually attend.

6. PLOS authors have the option to publish the peer review history of their article (what does this mean?). If published, this will include your full peer review and any attached files.

Reviewer #1: No

---

## [Author Response · Author response to Decision Letter 0]

4 Sep 2020

In general, the manuscript would benefit from English professional editing. There are many places with spelling issues, grammatical issues and other issues that can be addressed. 

We appreciate the feed back on this. We have previously sent it to a copyeditor for two rounds of editing and have sent it a third time to correct any potential grammatical issues.

Abstract 

The abstract need to include the results of the difference between Syrian refugees and local Jordanians at high risk of physical and mental health ailments. Such comparison needs to be also included in the results section, since it is well established that Syrian refugees have many physical and mental health issues due to experiencing multiple traumatic events of the war. However, Jordanians have not experienced such events and therefore such comparison is required. 

We have added this information in the abstract. Syrians did score significantly higher on post-traumatic stress symptoms, however, did not have differences on self-reported physical health. This is also presented in the tables, the results and the discussion.

Background 

The title needs to be changed. It is the Syrian Crisis and its impacts in Jordan, or Syrian refugees in Jordan. 

Thank you for your feedback on this. We have edited the title. Since we have a sample of Jordanians and Syrians we included Jordanians in the title. This paper is based on a larger study examining the impact of a health awareness intervention in Jordan. The reason we included both Jordanians and Syrians in the sample is because the project was done in a Ministry of Health clinic that served both non-camp Syrians and low-income Jordanians. It was therefore, unethical to not include Jordanians in the study and only offer the health awareness intervention to Syrians. Additionally, because this was done in a Ministry of Health clinic, there were restrictions from the Jordanian government to exclude Jordanians from the study. 

In the second paragraph, it is important to state that the number of Syrian refugees registered are by UNHCR in Jordan. Jordan has hosted more than 1.4 million Syrian refugees, who are not registered by UNHCR. 

We have added in the estimated unregistered Syrians to the paragraph.

Page 4: Please write the full term of PTS, since it can be confused with PTSD. We have changed this to represent clinical rates of Post-traumatic stress and spelled out the acronym. 

We recognize this may be confusing so changed the term throughout the manuscript to be PTSS rather than PTS symptoms.

Page 5: Forced migration, please change the term to forced displacement, since migration is associated with a choice. 

This has been changed

Page 6: the authors say “… which may be due to a lack of medications and appropriate health care, coupled with high risk for mental health issues that is caused by trauma these individuals may have experienced in their country of origin (33)”. Syrian refugees suffer from extreme poverty and insufficient resources that prevent them from seeking care. In Jordan, health care is private and very expensive. These consideration needs to be taken into account when considering the reasons to why Syrian refugees have high prevalence rates of NCDs.

We have added in the suggested literature expanding the discussion. We have also added in additional literature about the prevalence of NCDs in Jordanians as the paper is focusing on both Syrians and Jordanians. The rates of NCDs and co-morbid mental health symptoms is also high among Jordanians and we added the literature to account for this.

Social, physical, and mental health 

Please start this paragraph with the definition of social health.

We have changed the paragraph to begin with the definition of social health and also edited the section for clarity.

The background is a bit confusing, since it is focuses on the Syrian refugees and their physical and mental health. However, the authors states in page 7 that their study aims at examining the physical and mental health “within a Jordanian community near the Syrian border”. Where is the literature review related to the Jordanian community? 

Thank you for this insight. In revising the paper, we recognize that we did not have enough information in the background about physical and mental health difficulties among Jordanians. We have added a bit more of a discussion to include NCD and mental health statistics among Jordanians. Syrians do experience very high rates of NCDs and mental health difficulties, however, low-income Jordanians also experience high rates of NCDs and are disproportionately affected by mental health struggles. This is due to living in poverty with scarce resources. Additionally, our sample was collected in a community very close to the Syrian border, and our sample included Jordanians and Syrians

Methods 

Page 7: Please state that the fliers with information on the study were in Arabic. 

We have added this 

Page 8: Who were the research team trained by the PI? Were they Arabic speakers? Jordanians? Syrians? Both? Please specify their educational background as well since they may have affected the responses of participants. 

We have added the following: The research team included three Jordanian staff employed by the Royal Health Awareness Society (RHAS) and two ministry of health nurses all fluent in Arabic; two United States based Americares staff; and one university researcher.

Page 8: “self- report questionnaires” means that the participants have filled out the survey independently. However, according to the explanation provided, the research team helped participants with the survey. Please add ethical issues related to anonymous participation, confidentiality, incentives if provided, oral consent form or written consent form, etc. 

We have clarified this to: The questionnaire was administered through researcher assisted self-completion to ensure comprehension of the survey items. The ministry of health nurses and RHAS staff read the questions in Arabic, repeating and clarifying if the study participants expressed any difficulties with comprehension. 

We also have added additional information about the consenting procedures in this section.

Page 10: Usually the measures section starts with the description of demographics. Please relocate. 

We have added this to the beginning of the measures section

Page 11: How was the mediation effects tested? Were they also done via SPSS PROCESS? Please clarify. 

We conducted all analyses in SPSS 27 and re-located that information to the bottom of the statistical analysis section. 

Page 11: “We adjusted for the effect of sociodemographic correlates (age, gender, nationality, trauma exposure, and education level) in the moderation and mediation models.” Income needs to also be included with the controlled variables. If the authors found that all participants were low on income, they need to indicate that this was the reason to not include this variable as one of the controlled variables. Most of the sample made less than 500 JD a month (91.2%), therefore we excluded this as a control variable. 

Page 12: 

Please indicate results related to the comparison between Jordanians and Syrians, other than the obvious “Syrians reported significantly higher PTS scores than Jordanians”. If there were no other differences, please indicate that. 

We have indicated that there were no other differences beyond PTSS between Syrians and Jordanians. This is also reported in the abstract, table and discussion.

I also think that all the analyses need to be divided into Jordanians and Syrians to see the difference between the two groups, especially because of the large sample size. 

Thank you for the suggestion on the analysis. Our primary research question is to understand mediation and moderation mechanisms rather than comparing Jordanians versus Syrians. Thus, splitting the analysis would distract the results from our primary research goal. On the other hand, we agree that the reviewer raises an interesting further question on whether the effects differ by nationality, and we ran further models with an interaction term with nationality. None of the interaction terms are significant, and thus for better readability of the tables we retained the original format. 

Who are the “others” in the sample? Are they Palestinians? 

In table 1 we also edited nationality to represent other included Egyptian (n=2) and Palestinian (n=1)

Discussion 

Though the findings are important in the field of trauma studies, they lack addressing the specific population. Syrian refugees who were uprooted and displaced, naturally lack the social connections and interactions that Jordanians have. Therefore, the whole analyses need to address these differences both in the results section and the discussion. The authors only mention this comparison in page 14. 

Thank you for this insight. We did not find any differences between nationalities on social health, therefore, while the Syrians may naturally lack social connections, we are unable to assert this in the discussion. We did note the differences of social and physical health and PTSS in the results section between nationality and it is provided in Table 3. 

We do have a qualitative paper under review, however, that does discuss this in more detail. Since this paper is focusing on the mediating and moderating role of social health on the relationship of physical health and PTSS, we felt this may be outside of the scope of the paper.

Also, please add into the results and discussion what were the main traumatic events experienced by Jordanians, since for the Syrians, the war events are obvious. 

We have added a table (table 2) and described the specific trauma experiences among Syrians and Jordanians.

Page 15: Gender differences in PTSD among Syrian refugees can also be found in: Rizkalla N, Segal SP. War can harm intimacy: consequences for refugees who escaped Syria. J Glob Health. 2019; 9(2):1–10. Available from: https://doi.org/10.7189/jogh.09.020407

We have added some additional information and the citation from this article.

---

## [Editor Report · Decision Letter 1]

8 Oct 2020

Post-traumatic stress, social, and physical health: A mediation and moderation analysis of Syrian refugees and Jordanians in a border community

PONE-D-20-11520R1

Dear Dr. Powell,

We’re pleased to inform you that your manuscript has been judged scientifically suitable for publication and will be formally accepted for publication once it meets all outstanding technical requirements.

Kind regards,

Neil J. Vincent, Ph.D.

Academic Editor

PLOS ONE
---

## [Editor Report · Acceptance letter]

15 Oct 2020

PONE-D-20-11520R1 

Post-traumatic stress, social, and physical health: A mediation and moderation analysis of Syrian refugees and Jordanians in a border community 

Dear Dr. Powell:

I'm pleased to inform you that your manuscript has been deemed suitable for publication in PLOS ONE. Congratulations! Your manuscript is now with our production department. 

Kind regards, 

on behalf of

Dr. Neil J. Vincent 

Academic Editor

PLOS ONE